

# Using Remote Sensing to Monitor the Spring Phenology of Acadia National Park across Elevational Gradients

Yan Liu[1,2], Caitlin McDonough MacKenzie[3,4], Richard B. Primack[4], Michael J. Hill[5,6], Xiaoyang Zhang[7], Zhuosen Wang[8,9], Crystal B. Schaaf[1]

[1]School for the Environment, University of Massachusetts Boston, Boston, 02125, USA
[2]Aerospace Information Research Institute, Chinese Academy of Sciences, Beijing, 100094, China
[3]Climate Change Institute, University of Maine, Orono, 04469, USA
[4]Department of Biology, Boston University, Boston, 02215, USA
[5]College of Science and Engineering, Flinders University, Adelaide, 5042, Australia
[6]Department of Earth System Science and Policy, University of North Dakota, Grand Forks, 58202, USA
[7]Geospatial Sciences Center of Excellence, South Dakota State University, Brookings, 57007, USA
[8]Earth System Science Interdisciplinary Center, University of Maryland, College Park, 20742, USA
[9]Terrestrial Information Systems Laboratory, NASA Goddard Space Flight Center, Greenbelt, 20771, USA

*Correspondence to:* Yan Liu (liuyan@radi.ac.cn) and Crystal B. Schaaf (crystal.schaaf@umb.edu)

**Abstract.** Greenup dates of the mountainous Acadia National Park, were monitored using remote sensing data (including Landsat 8 surface reflectances (at a 30 m spatial resolution) and VIIRS reflectances adjusted to a nadir view (gridded at a 500 m spatial resolution)) during the 2013-2016 growing seasons. Ground-level leaf-out monitoring in the areas alongside the north-south-oriented hiking trails on three of the park's tallest mountains (466 m, 418 m, and 380 m) was used to evaluate satellite derived greenup dates in this study. While the 30 m resolution would be expected to provide a better scale for phenology detection in this mountainous region than the 500 m resolution, the daily temporal resolution of the 500 m data would be expected to offer vastly superior monitoring of the rapid variations experienced during vegetation greenup along elevational gradients. Therefore, the greenup dates derived from the Landsat 8 Enhanced Vegetation Index (EVI) data, augmented with Spatial and Temporal Adaptive Reflectance Fusion Model (STARFM) simulated EVI values, does provide more spatial details than VIIRS data alone and agree well with field monitored leaf out dates. Satellite derived greenup dates from the 30 m of Acadia National Park vary among different elevational zones, although the date of greenup is not always the most advanced at the lowest elevation. This indicates that the spring phenology is not only determined by microclimates associated with different elevations in this mountainous area, but is also possibly affected by the species mixture, localized temperatures, and other factors in Acadia.

## 1. Introduction

Spring phenology is defined as the timing of physiological and morphological changes in vegetation that herald the start of new growth and are characterized by the timing of leaf out, bud burst, and flowering (Barr et al., 2009; Rechid et al., 2009; Richardson et al., 2010; Song, 1999). Changes in phenology are sensitive indicators of ecosystem response to climate change



and such changes provide important feedbacks to the climate system, through albedo, nutrient cycling, and so on (Morisette et al., 2009). Environmental factors that influence spring phenology, such as temperature, sunlight, and precipitation, tend to vary widely in mountainous regions due to the influence of complex topography (Beniston, 2003), leading to a variety of diverse niches within the broader ecosystem (Bales et al., 2006; Haslett, 2006). Thus the study of vegetation phenology in these

topographically complex mountainous areas may provide early indications of changes in ecosystem processes resulting from climate change (Gallinat et al., 2015; Morisette et al., 2009).

Vegetation phenology has been historically monitored at the ground level though field surveys. This strategy focuses on species centered phenophases (for particular plant species), and such records have been collected over decades and even centuries in some locations (Ge et al., 2003; Hameed and Gong, 1994; Schwartz, 2013). This approach is still widely used today to record

phenological transitions in various ecosystems, including forest, savanna, grassland, and agriculture (Dickinson et al., 2010; Ma et al., 2007, 2016; Miller-Rushing and Primack, 2008; Panchen et al., 2015; Polgar et al., 2014). While ground level phenological observations are straightforward to measure, manual monitoring can be quite laborious and time consuming, especially in complex ecosystems or in mountainous regions where sampling is often constrained to small areas.

In recent decades, satellite imagery has been increasingly used to capture Land Surface Phenology (LSP) over various

ecosystems worldwide (Friedl et al., 2010; Ganguly et al., 2010; Reed et al., 1994; Zhang et al., 2012) with studies focused on mountainous areas in North America (Hudson Dunn and de Beurs, 2011; Hwang et al., 2014), in China (Qiu et al., 2013; Zeng et al., 2010), and in Europe (Fontana et al., 2008; Guyon et al., 2011). LSP describes the overall phenological dynamics of the vegetation communities within a satellite sensor field of view (de Beurs and Henebry, 2004), and have generally used relatively coarse resolution imagery (250 m -1000 m),   from sensors including the Advanced Very High Resolution Radiometer

(AVHRR), the Moderate Resolution Imaging Spectroradiometer (MODIS), the Visible Infrared Imaging Radiometer Suite (VIIRS), and Satellite Pour l'Observation de la Terre (SPOT)-VEGETATION sensors for phenological monitoring, as these sensors can provide daily or near-daily observations worldwide (Fontana et al., 2008; Friedl et al., 2010; Zhang et al., 2012, 2018; Zhao et al., 2015). However, in mountainous environments, where the elevation vary significantly within one pixel, these sensors may not provide the best spatial scale to describe the dynamics of the vegetation phenology. Imagery from

satellites such as Landsat, HuanJing (HJ), Sentinel 2, and GaoFen (GF), provides higher spatial resolutions and is more appropriate to measure vegetation phenology in heterogeneous regions (Fisher et al., 2006; Melaas et al., 2013; Pan et al., 2015; Walker et al., 2012). However, due to the relatively low temporal frequency of these sensors, and the notoriously cloudy conditions in mountainous regions, these higher resolution sensors often only acquire a few cloud-free images over an entire growing season. Efforts have been made to improve upon the temporal resolution of Landsat and other higher resolution

imagers by using multiple-year aggregation methods (Avitabile et al. 2012; Melaas, Friedl, and Zhu 2013; Sulla-Menashe, Friedl, and Woodcock 2016) and data fusion algorithms, such as the Spatial and Temporal Adaptive Reflectance Fusion Model (STARFM), the Enhanced STARFM (ESTARFM), and the Spatial Temporal Adaptive Algorithm for mapping Reflectance Change (STAARCH) (Gao et al., 2006; Hilker et al., 2009; Zhu et al., 2010).



In this study, Landsat and VIIRS satellite data were used to investigate the variation in the spring phenology of vegetation along elevation gradients in Acadia National Park in the northeastern United States. MODIS daily Nadir Bidirectional Reflectance Distribution Function Adjusted Reflectance (MCD43, V006) were used as well. Acadia National Park was created in 1916 to conserve the natural environment of coastal Maine and minimize direct human impacts (Shafer, 1999). A great deal of environmental monitoring has been carried out at Acadia to document biodiversity, species abundance (Burns et al., 2003; O'Connell et al., 2004), phenology (Klosterman et al., 2014; Miller-Rushing et al., 2011; Monahan et al., 2016), and air quality (Sofuoglu et al., 2013). In particular, three field transects across elevational gradients have been established at Acadia, to specifically record the spring vegetation phenology from 2013 to 2016 (McDonough MacKenzie et al., 2019). These meticulously collected field data provide an ideal set of ground observations to assess the ability of satellite derived products to capture the greenup dates of this topographically complex area of Acadia National Park. The objectives of this study are:

1. To study the variations in greenup dates detected at different spatial scales; and

2. To examine the differences between field observed leaf out dates and satellite detected greenup dates;

3. To analyze the variation of these satellite derived greenup dates among different elevational zones and land covers.

## 2. Materials and methods

### 2.1. Research area

Acadia National Park is made up of areas of Mount Desert Island, the Schoodic Peninsula, and other nearby offshore islands along the coast of Maine. The research here is focused on the Mount Desert Island unit of the park, which is mostly covered by evergreen forest, mixed forest, wetlands, and shrub/scrub (Fig. 1). This topographically complex area contains the tallest coastal mountain on the North Atlantic seaboard, Cadillac Mountain with an altitude of 466 m.

This area experiences relatively long and cold winters, and abundant precipitation. As recorded at the McFarland Hill weather station (elevation of 158 m) located within the park, the minimum and maximum monthly air temperature ranged from -16 to 26 Celsius degrees, and the minimum and maximum monthly precipitation ranged from 25 to 205 mm during 2013 to 2016 (Fig. 2). Among those years, the temperatures of the springtime months displayed strong yearly variations: the coldest month of March was in 2014, and the warmest month of May was in 2015 (Fig. 3).

### 2.2. Satellite data

The satellite data used for LSP detection in this study included the gridded 500 m daily Nadir Bidirectional Reflectance Distribution Function (BRDF) Adjusted Reflectance (NBAR) from the NASA operational VIIRS products (Liu et al., 2017), and the 30 m atmospherically corrected surface reflectances from Landsat 8 Operational Land Imager (OLI) (Vermote et al., 2016).

The 500 m VIIRS NBAR products are part of the VIIRS BRDF, Albedo, and NBAR product packages (VNP43I), which are available on a daily time step, and thus are able to capture rapid temporal changes (Liu et al., 2017). The VNP43I products,





starting with the first launch of VIIRS in 2012, on board the Suomi-NPP satellite, have been developed in order to produce a continuous record equivalent to that from the MODIS record (Schaaf et al., 2002, 2011; Wang et al., 2018). The continuous NBAR products (VNP43I4) provide reflectance values with consistent nadir viewing geometry that removes the variations in the directional surface reflectances caused by the viewing geometry variation of VIIRS (±56° across-track) (Liu et al., 2017).

In order to provide solar illumination geometry consistent with the Landsat acquisitions, the solar zenith angle of the Landsat overpass time was used to calculate the VIIRS NBARs from the BRDF model parameter products (VNP43I1) for this study. Landsat 8 was launched in February 2013 extending the long term record of the Landsat program (Roy et al., 2014; Wang et al., 2016). The atmospherically corrected surface reflectance products from Landsat 8 OLI have higher spatial resolutions (30 m), but lower temporal resolutions (a 16 day repeat cycle) as compared to the daily VIIRS 500 m gridded data. Fortunately,

Acadia National Park falls in the overlap zone between Landsat path 11/row 29 and Landsat path 10/row 29. Therefore the number of possible Landsat observations is doubled. Landsat 8 OLI imagery are acquired with near nadir viewing geometry (only up to 7.5° off nadir) (Roy et al. 2016), thus atmospherically corrected surface reflectances, downloaded from EarthExplorer, are used in this study directly, without any additional angular correction.

## 2.3. LSP detection

Piecewise logistic functions (Zhang, 2015; Zhang et al., 2003) are used to detect the LSP from the satellite derived Enhanced Vegetation Indices (EVI) time series. EVI values have been established to reduce residual atmospheric noise and these values do not saturate at high vegetation densities (Huete et al., 2002). Four phenology transition dates (greenup, maturity, senescence, and dormancy) can be derived using the piecewise logistic method. The greenup dates, defined as the onset of the EVI increase, are used in this study of Acadia National Park on Mount Desert Island. However, since the 500 m VIIRS products do not

include blue band spectral information, a two band variant, the EVI2, was substituted for EVI (Jiang et al., 2008). Therefore, the Landsat 8 EVI is calculated using Eq. (1) and the VIIRS EVI2 is calculated using Eq. (2).

$$EVI = 2.5 * \frac{\rho_{NIR} - \rho_R}{\rho_{NIR} + 6*\rho_R - 7.5\rho_B + 2.5}, \tag{1}$$

$$EVI2 = 2.5 * \frac{\rho_{NIR} - \rho_R}{\rho_{NIR} + 2.4*\rho_R + 1}, \tag{2}$$

where $\rho_R$, $\rho_{NIR}$, and $\rho_B$ represent the reflectance of the red, Near InfraRed (NIR), and blue bands respectively.

The daily resolution of the VIIRS NBAR product meets the temporal requirement for phenology monitoring, although, the 16-day (or even 8-day in the overlap region) temporal resolution of Landsat 8 surface reflectance does not. Therefore, Landsat 8 OLI EVI was fused with daily MODIS NBAR EVI using the STARFM (Gao et al., 2006) algorithm to improve the temporal resolution of 30 m imagery. This fusion model utilizes a pair of coarse resolution and fine resolution images obtained at the same time and a subsequent coarse resolution image at a prediction time to estimate a corresponding fine resolution image.

With the establishment of satellite derived EVI time series, piecewise logistic functions (Zhang, 2015; Zhang et al., 2003) were fitted to the various time series to compute the greenup dates of Acadia National Park.



### 2.4. Field observed phenology along the hiking trails

Three north-south field transects have been established along hiking trails on Cadillac, Pemetic, and Sargent mountains in Acadia (McDonough MacKenzie et al., 2019) to record the leaf out and flowering date of about thirty plant species, including herbs, shrubs, and small trees from 2013 to 2016 (Fig. 4). This "trail-as-transects" approach was originally designed as a citizen
science program, therefore it represents an easy approach for hikers to collect data while they hike the mountain trails (McDonough MacKenzie et al., 2019).

For the monitoring of this project, each hiking trail was separated into different elevational zones based on aspect (North and South) and elevation (low, medium, high, and summit). The altitude range of each elevational zone is standardized across these Acadia mountains and aspects; "low" is below 183 m, "medium" 183-274 m, "high" 274-366 m, and "summit" above 366 m
(Fig. 4). Thus, 24 monitoring zones were established. The hiking trail of Giant Slide is connected with the north hiking trail of Sargent Ridge. The north and south hiking trails of Cadillac Ridge are not connected because there is a parking lot at the summit of Cadillac Mountain.

In each elevational zone, the phenophase of understory species and of overstory deciduous trees along the hiking trail were monitored twice a week from April 1 through June 30. Note that the evergreen trees in these regions were not monitored. The
leaf out date of each species was defined as the first observed leaf out date of any plant for that species in each elevational zone along the hiking trails. This approach is similar with the method of the USA National Phenology Network (Denny et al., 2014). In addition, within each monitoring zone, each species was also categorized as "Common", "Occasional", and "Uncommon" based on the abundance occurring in that monitoring zone. Among all the elevational zones, no field observed species are classified as a common species in the low and medium elevational zones of the Pemetic North and South trails, as
these zones are densely covered by a spruce/fir forest with little or no understory. More details about the field observation can be found in the supplemental material.

### 2.5. Pixel species composition survey

The hiking trails represent a much smaller area than that of a 30 m pixel let alone a 500 m pixel (Fig. 4), thus the species composition along the hiking trails may be quite different than the species composition of larger pixels that cover the trails
and adjoining regions. Therefore the 30 m or 500 m pixels are generally heterogeneously covered with an assortment overstory and understory species. Furthermore, the species composition of 500 m pixels and even 30 m pixels covering the hiking trails is difficult to observe due to the complexity of the topography.

There are four 30 m pixels which are possible to access and are covered with relatively homogeneous vegetation in the high elevational zone on the north side of Cadillac Mountain. Although these pixels are more accessible than others, it's still
impossible to perform a stem by stem species surveys in the field due to the topographic variation. A general species survey was performed for these pixels by knowing the corners' locations of these pixels. These four pixels have the same mix of



species: the dominant overstory species is grey birch (Betula populifolia), and the dominant understory species are bunchberry (Chamaepericlymenum canadensis) and wild raisin (Viburnum nudum).

## 2.6. Analysis of greenup dates

The greenup dates as detected by satellites are analyzed through the following steps:

1. Landsat estimated greenup dates are compared with field monitored first leaf out dates through two approaches. One is a direct comparison of the dominant species over the relatively homogeneous 30 m pixels located on the North Cadillac high elevational zone. Another approach considers each elevational zone as a separate community. In each zone, first leaf out dates of all common species monitored in the field are averaged into a field monitored community leaf out date; the greenup dates of all Landsat pixels, excluding predominantly evergreen forest pixels, are then also averaged to produce the Landsat greenup

date of the community. Then the Landsat greenup date of the community is compared with corresponding field monitored community leaf out date. The Landsat evergreen forest pixels were identified using the 30 m National Land Cover Database (NLCD) of 2011;

2. The variations in greenup dates at the 30 m scale across elevational gradients of the whole park and for each landcover were analyzed;

In addition with regards to the STARFM derived EVI time series, the differences in greenup dates between the 30 m scale and the 500 m scale were also evaluated.

## 3. Results

### 3.1. Detailed evaluation of the STARFM results over Acadia National Park

In addition to the comparisons with field data, the STARFM simulations were also compared to a few pairs of Landsat images

from 2013 to 2016 (Table 1). This comparison indicates that the STARFM simulated EVIs are highly correlated with the real Landsat 8 EVIs. As displayed in the scatter plots (Fig. 5), the points fall around the 1:1 line, and the majority points fall within the ±0.025 boundary, with coefficients (R) higher than 0.9, RMSEs (Root Mean Squared Error) smaller than 0.05, and absolute biases smaller than 0.04 between STARFM simulated and real Landsat 8 EVIs. This accuracy assures that the STARFM simulated results can be used to augment the real Landsat 8 images for phenology monitoring. The residual scatter in the figure

appears to be primarily due to uncertainties brought by the use of STARFM algorithm and possible residual atmospheric effects in the atmospheric correction of Landsat 8 images.

### 3.2. Satellite detected greenup dates over the Acadia National Park on Mount Desert Island

The greenup date maps as detected by the VIIRS EVI2 and by the Landsat 8 EVI, augmented with the simulated STARFM (Landsat 8 + STARFM) EVI are displayed in Fig. 6 and 7. At 30 m, obviously more spatial details are captured than at the 500

m scale in this topographically complicated area.



### 3.3. Comparison of the Landsat detected greenup dates with the field recorded first leaf out dates

#### 3.3.1. Greenup in homogeneous pixels

As the dominant overstory species of the relatively homogeneous pixels, grey birch, was only monitored from 2014 to 2016 in the field, the comparison between the field observations and satellite detections is focused on these three years. The greenup

dates detected with the Landsat 8 + STARFM EVI time series agree well with the field observed leaf out dates (bias = 0, RMSE = 2) (Fig. 8) over these pixels.

As displayed in Fig. 8, the annual variation between field observation and satellite detection also has similar trends. In year 2015 there were earlier leaf out and greenup dates than in 2016 and 2014. Greenup and leaf out dates in these pixels are associated closely with corresponding mean May temperatures: both greenup and leaf out occur earlier in years with warmer

Mays. The mean May temperature in 2015 was higher than that of 2014 and 2016 by about 1.9° C and 1.3° C respectively (Fig. 4) and the leaf out dates of these pixels, as monitored in the field for 2015, were 8 days and 2 days earlier than the leaf out dates for 2014 and 2016, respectively. Accordingly, the greenup dates of 2015 detected using the Landsat 8 + STARFM EVI were 7 days and 5 days earlier than the dates detected for 2014 and 2016, respectively (Fig. 8).

#### 3.3.2. Over each elevational zone

The greenup dates of each elevational zone from 2013 to 2016, as detected using time series of the Landsat 8 + STARFM EVI, were further compared with field observed leaf out dates. The elevational zones covered by evergreen forest and developed areas, such as houses and roads (the medium elevational zone of the Sargent North trail, the low and medium elevation zones of the Pemetic North and South trails) are not analyzed here.

Over the hiking trails of these three mountains, satellite derived community greenup dates are generally close to the field

observed community leaf out dates with a small bias (-2 days) and RMSE (10 days) (Fig. 9). However, the divergence between field observation and satellite detection is more significant over most of the trails than is associated with the relatively homogeneous pixels on Cadillac North trail. Most likely this is due to more diverse species composition found along these hiking trails even within a relatively small Landsat 30m pixel over complex terrain.

### 3.4. Phenology variation across elevational gradients in Acadia National Park

Variation in the greenup dates among different elevational zones from 2013 to 2016 are displayed as boxplots in Fig. 10. Greenup and elevation are well correlated in 2013, then the low elevational zone had the earliest greenup dates and the summit zone had the latest. The medium value of the greenup dates of each elevational zone (the red central mark of the boxplot in figures) advances 1.3 days on average as one moves to lower elevations. However, in 2014-2016, the relationship between greenup dates and elevational zones is less clear. In these years, the lowest elevation zones do not consistently show the earliest

greenup dates, and the phenology appears to be decoupled from the elevation gradient for these communities.



Detailed analysis of the variation in greenup dates was performed for each land cover type among the different elevational zones. In mixed forests, the date of the greenup is correlated with elevation zone (earlier greenup at lower elevations) for all four years. Similar trends were found in evergreen forest in 2014, 2015, and 2016 and the deciduous forest in 2013 and 2014. Herbaceous cover also generally reflected later greenup dates at higher elevations. However herbaceous cover in the summit

zone deviates from this pattern; here medium greenup occurs 2 days before the high elevation zone. In summary, the trends of greenup dates along the elevational zones are different for each landcovers, and these seem to also diverge from the general trends of the entire park.

## 4. Discussion

### 4.1. Greenup dates as detected by different satellite sensors

The differences in the satellite detected greenup dates between 30 m and 500 m observations appear to be mainly caused by spatial differences, as more spatial detail can be captured by the 30 m data than from the 500 m. The spectral wavelengths of the 500 m VIIRS NBAR values and of the 30 m Landsat 8 surface reflectance are not identical (Liu et al., 2017). Therefore it's possible that the spectral differences will also add uncertainties to phenology monitoring, but the influence is expected to be less than the effect of the spatial differences.

STARFM was used to augment the temporal resolution for Landsat 8, and the derived greenup dates appear to agree with the ground measured leaf out dates. However it must be recognized that the STARFM results are simulated values and not values truly observed by the higher spatial resolution satellite sensors, and this process can introduce its own uncertainties (including the spectral differences between the coarse resolution MODIS and fine resolution Landsat 8 images, the temporal variation of inputs and prediction dates, geolocation errors, and even residual effects from the viewing and solar angular corrections (Gao

et al., 2006)). In the future, another approach to improve the temporal resolution may be to combine images from a number of sensors that have similar higher spatial resolution designs, such as Landsat, HJ, and Sentinel 2. However, the spectral differences, calibration accuracies, and position accuracy differences between all of these sensors may bring additional uncertainties to the time series needed for phenology monitoring.

### 4.2. Factors causing the differences between the field observed leaf out dates and Landsat scale detected greenup dates

Comparison of the 30 m greenup dates detected from the Landsat 8 + STARFM values with the field observed leaf out dates illustrated a relatively good agreement (bias of 0 and RMSE of 2) over the relatively homogeneous pixels. However, the agreement of field observations and Landsat detections over all the elevational zones along the hiking trails is less ideal (bias of -2 and RMSE of 10) compared with over the relatively homogeneous pixels. The field observation methods focused on collecting data along hiking trails, which represent a much smaller area than that captured by a 30 m Landsat pixel. The species

composition varies significantly within the park, thus the species composition along the hiking trails may be quite different



than the species composition of an entire Landsat pixel that covers and adjoins the trails. This sampling bias could contribute to the differences seen between the field observed leaf out dates and the satellite detected greenup dates.

### 4.3. Phenology difference for different elevational zones.

Elevation has been shown to affect the greenup dates over the entire park on the Mount Desert Island as detected at 30 m. However, elevational zones with higher altitude do not always correspond with later greenup dates, and the elevational trend in green up varies with landcover and across different years. The Park's heterogeneous species composition across elevation zones and within landcover types appears to complicate the elevational trends, as so does local microclimates.

### 5. Conclusion

This study documented elevational variation in greenup dates in Acadia National Park, while leveraging field observed phenology to validate remote sensing monitoring over a topographically complex area.

The greenup dates derived from satellite at 30 m scale agreed well with the field monitored first leaf out dates. The low biases and RMSEs between field observed and satellite monitored phenology assure that, although it's difficult to monitor species centered phenology using satellite, the overall phenological dynamics detected by satellite can be used to fill the gap for field observation especially in topographically complex areas that are hard to reach in the field.

At 30 m scale, more phenology details can be detected than at the more commonly used coarse resolutions of satellite data. We found variation in greenup dates across the elevation zones in Acadia, although the relationship between greenup dates and elevation changes among years and landcovers. The species variation in each elevational zone and the possible varied response of these landcover to local changes in meteorological factors changes appear to contribute to this variation. For this study, data fusion (of Landsat and MODIS) was utilized to improve the temporal resolution of the time series required for phenology monitoring. In the future, higher resolution images from other platforms, such as Sentinel 2 A/B, HJ, and GF, may also be incorporated to produce higher spatial and temporal resolution time series for phenological monitoring. With the availability of these higher resolution satellite data, phenological monitoring of even topographically complex areas may become more routine in the future.

**Data availability.** The MODIS and VIIRS data can be accessed through the NASA Level-1 and Atmosphere Archive & Distribution System (LAADS) (https://ladsweb.modaps.eosdis.nasa.gov/). The Landsat data can be accessed through USGS EarthExplorer (https://earthexplorer.usgs.gov/). The ground measured data is available from the Dryad Digital Repository (https://doi.org/10.5061/dryad.1n93p40). The 30 m ASTER DEM can be accessed through NASA Earthdata Search (https://search.earthdata.nasa.gov). The 30 m National Land Cover Database (NLCD) can be accessed through USGS Earth Resources Observation and Science (EROS) Center (https://www.usgs.gov/centers/eros/science/national-land-cover-database). The shapefile of Acadia National Park can be downloaded from National Park Service Geographic Information Systems Data



and Information (https://www.nps.gov/gis/data_info).

**Author contributions.** Yan Liu designed the study and wrote the paper with contributions from all co-authors. This study was initiated from the collaboration between Richard Primack and Crystal Schaaf. Caitlin McDonough Mackenzie led the ground-level spring phenology monitoring, with Richard Primack, Crystal Schaaf, and Yan Liu participated. Xiaoyang Zhang performed Piecewise logistic functions to the satellite derived Vegetation Indices time series.

**Acknowledgements.** The MODIS and VIIRS processing has been supported by NASA grants NNX14AI73G and NNX14AQ18A, and the Landsat processing has been supported by USGS award G12PC00072. The authors would like to thank Abraham J. Miller-Rushing, Acadia National Park, U.S. National Park Service, for his support on providing thoughts and data access.

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



**Table 1. Dates of Acadia National Park Landsat 8 and MODIS images used for evaluating the STARFM simulation.**

| Date of input images | Date of simulated and testing images |
| --- | --- |
| Year 2013 DOY 227 | Year 2013 DOY 236 |
| Year 2014 DOY 255 | Year 2014 DOY 262 |
| Year 2015 DOY 194 | Year 2015 DOY 210 |
| Year 2016 DOY 101 | Year 2016 DOY 133 |

The STARFM results for the dates listed in the right column were simulated using paired Landsat 8 and MODIS images of the dates listed in the left column and the MODIS images of the dates listed in the right column. The Landsat 8 images of the dates listed in the right column were used as the real observations to evaluated the STARFM simulated values.





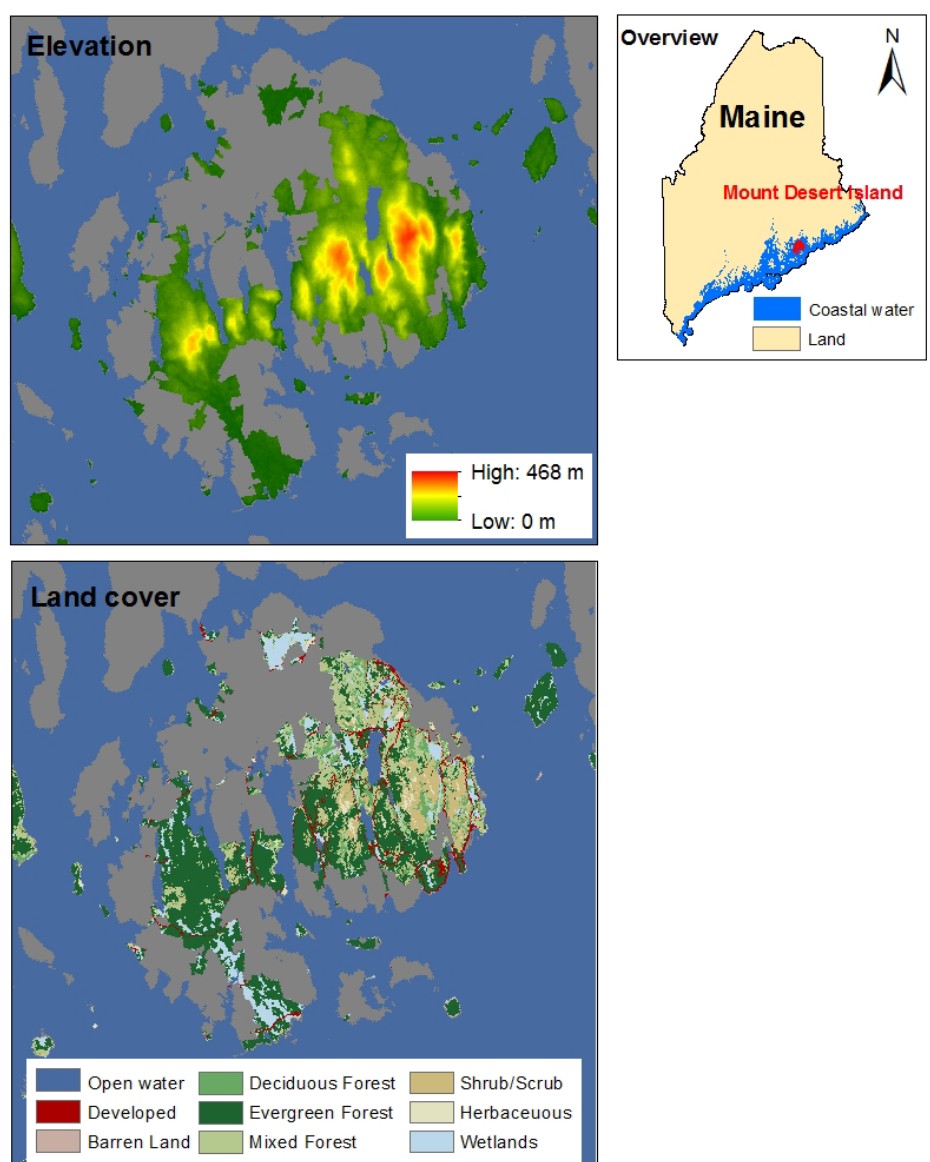

**Figure 1. Elevation and landcover of Acadia National Park on Mount Desert Island. Grey indicates the area doesn't belong to the Park and not analyzed in this study. The Elevation is from the 30 m ASTER DEM. The landcover is from the 30 m National Land Cover Database (NLCD) of 2011.**



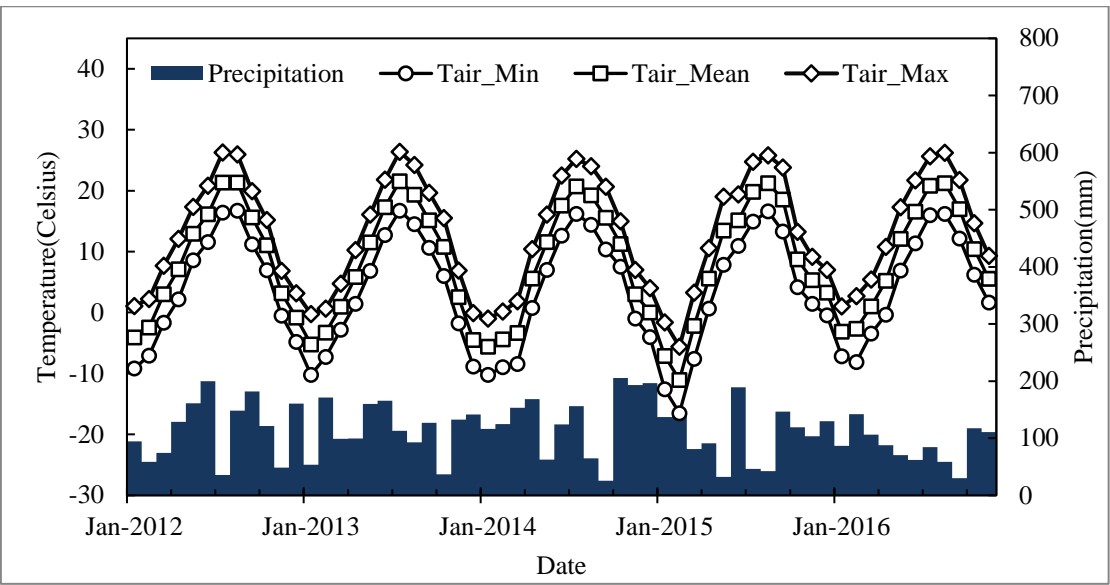

**Figure 2. Temperature and precipitation values recorded at McFarland Hill weather station in Acadia National Park. Tair_Min, Tair_Mean, and Tair_Max represent minimum, mean, and maximum air temperature of each month.**

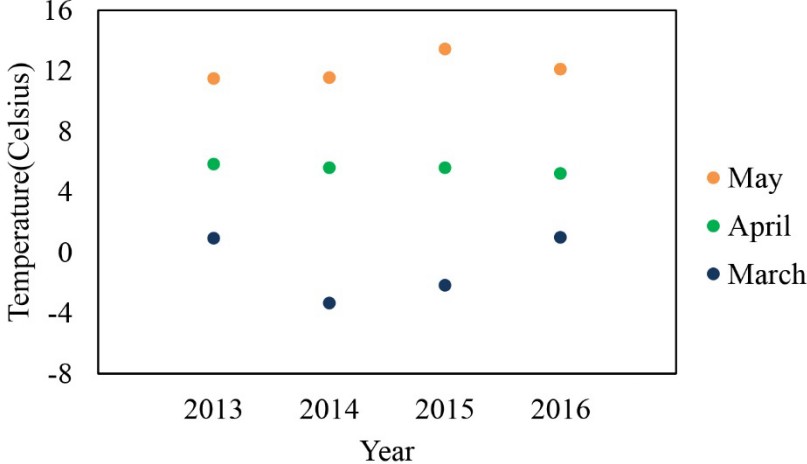

**Figure 3. Spring Monthly temperatures recorded at McFarland Hill weather station in Acadia National Park.**



**Figure 4.  Elevational zones on the hiking trails.**





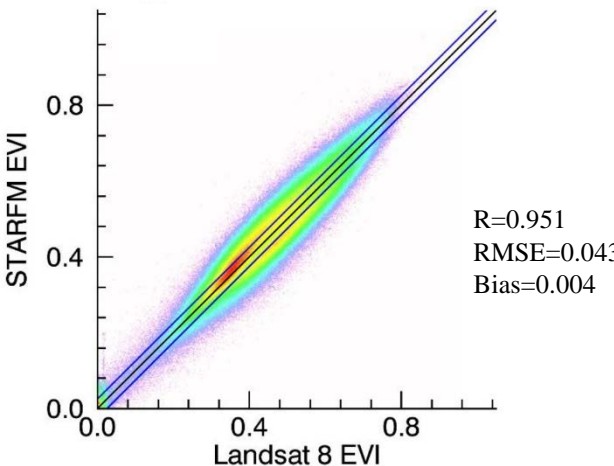

**Figure 5. Density plots of the STARFM results vs. the corresponding Landsat 8 EVI over Acadia National Park. Red indicates higher density and light purple indicates lower density. Central lines are the 1:1 lines, and outer lines are the 0.025 offset lines.**





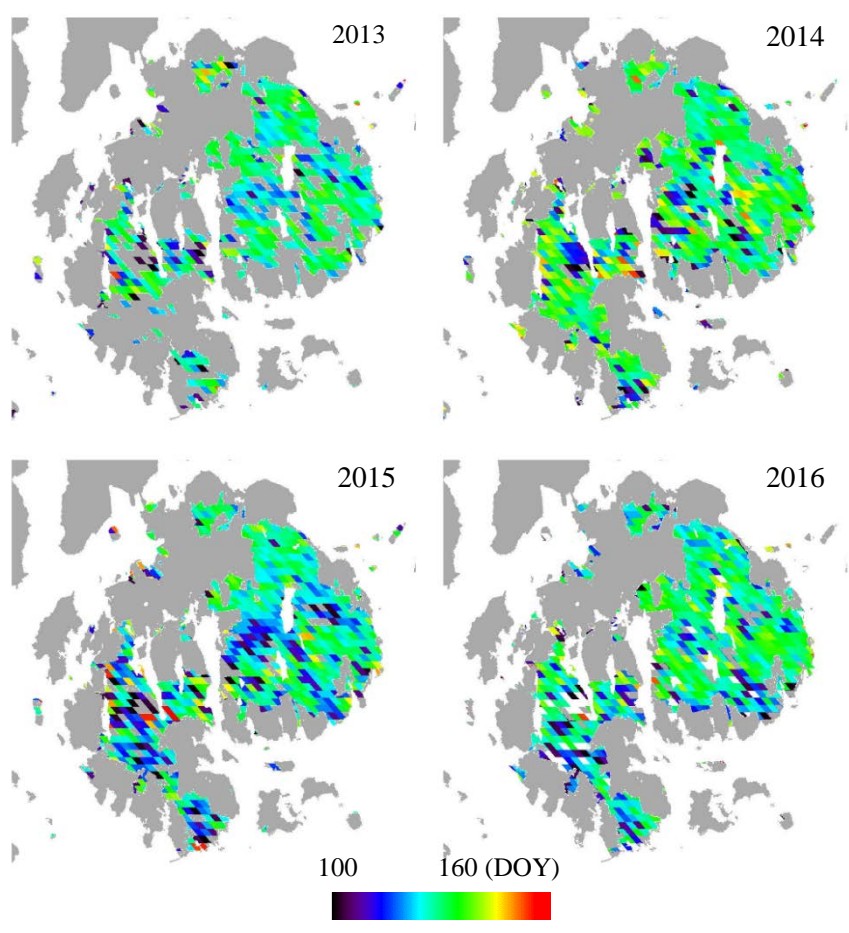

**Figure 6. The greenup dates of Acadia National Park on Mount Desert Island detected by VIIRS EVI2 from 2013 to 2016. Grey indicates the area doesn't belong to the Park and not analyzed in this study.**





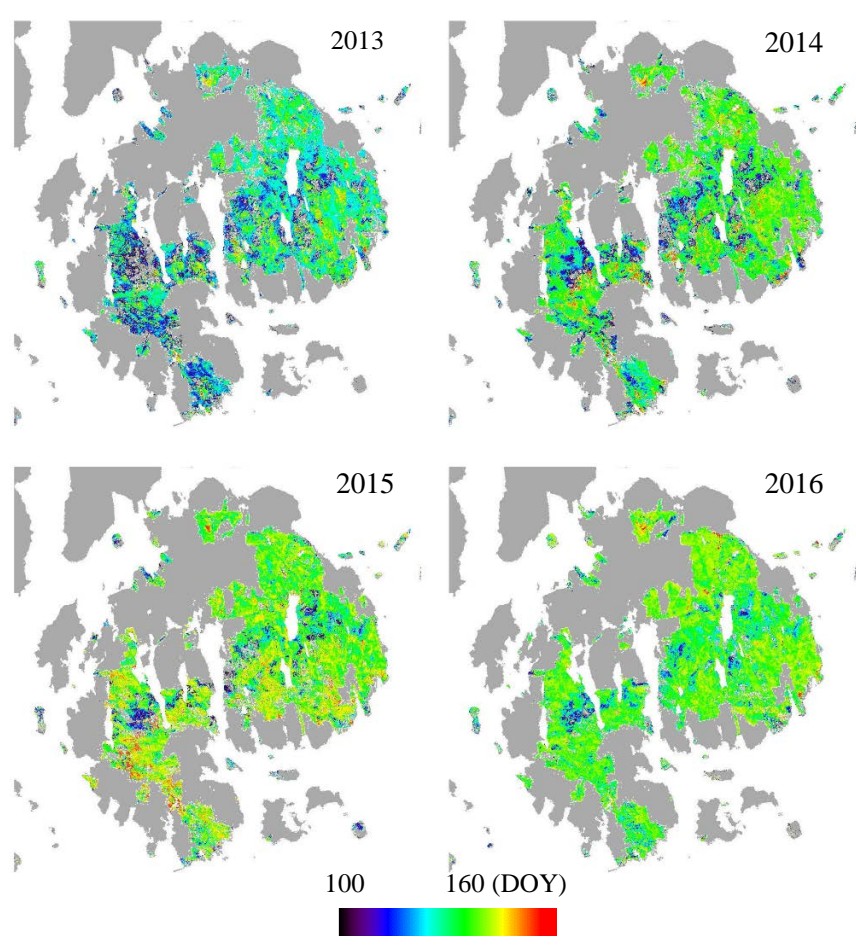

**Figure 7. The greenup dates of Acadia National Park on Mount Desert Island. Detected by the Landsat 8 EVI augmented with simulated STARFM EVI from 2013 to 2016. Grey indicates the area doesn't belong to the Park and not analyzed in this study.**



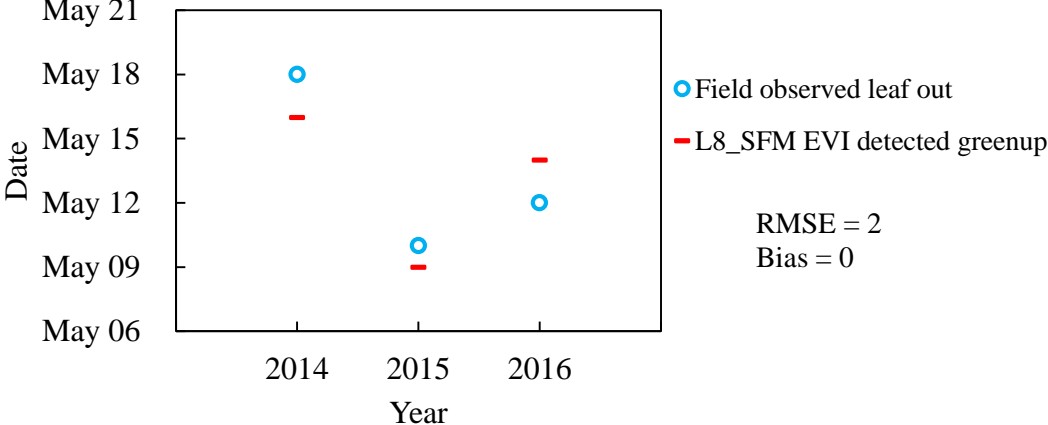

**Figure 8. Annual variation in spring phenology dates of the 30 m homogeneous pixels. L8_SFM indicates the Landsat 8 augmented with simulated STARFM results.**

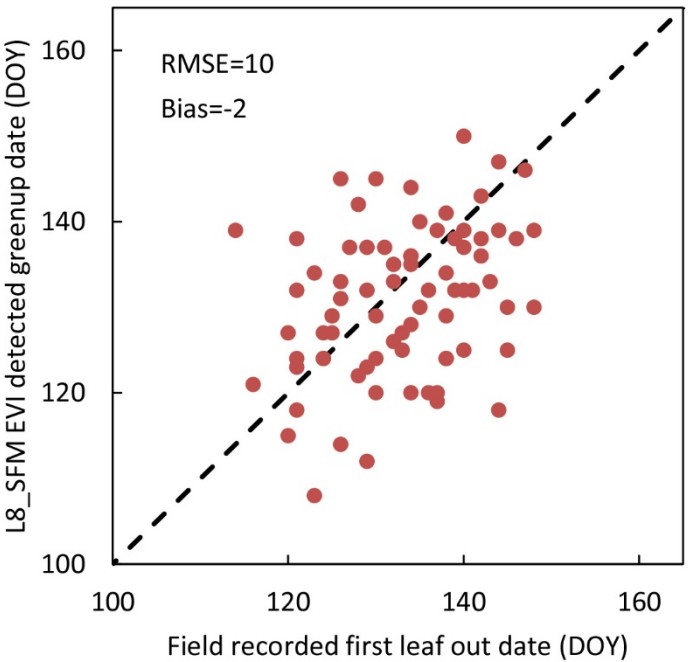

**Figure 9. Scatter plot of the Landsat scale detected greenup date and the field observed first leaf out date of each community over all hiking trails from 2013 to 2016. L8_SFM indicates Landsat 8 + STARFM.**



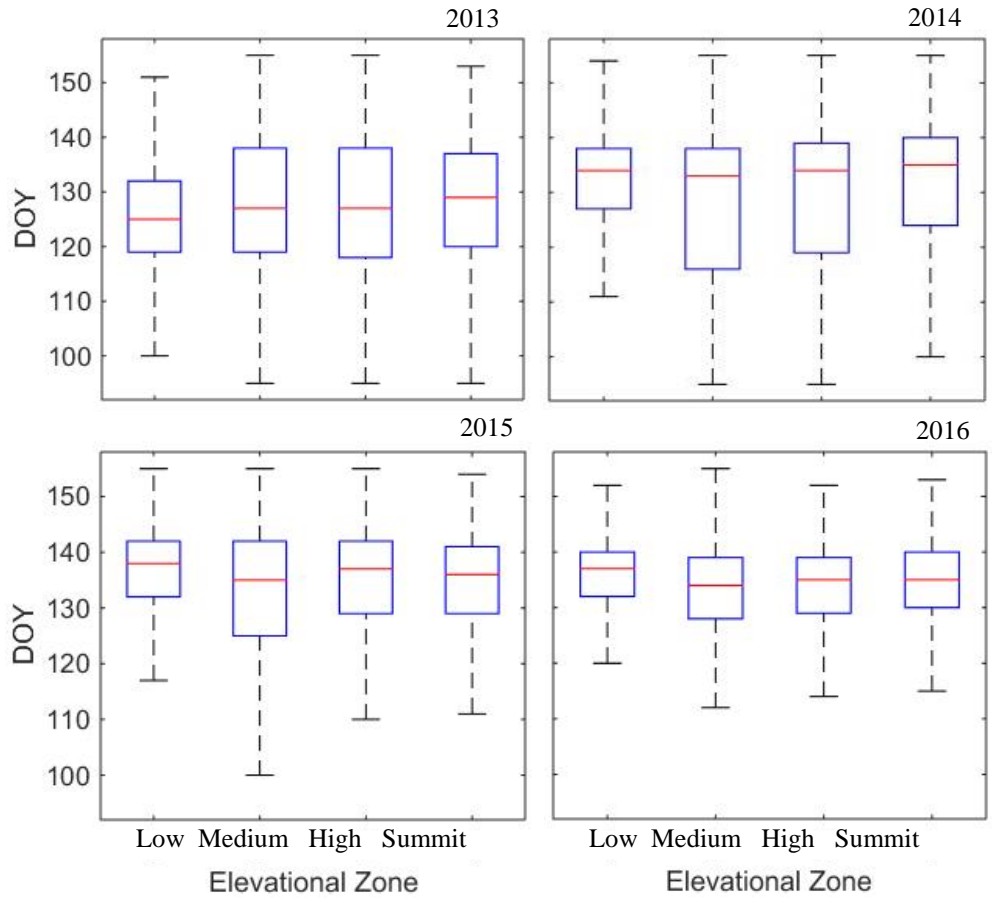

**Figure 10. Variation in Greenup dates among different elevational zones from 2013 to 2016**





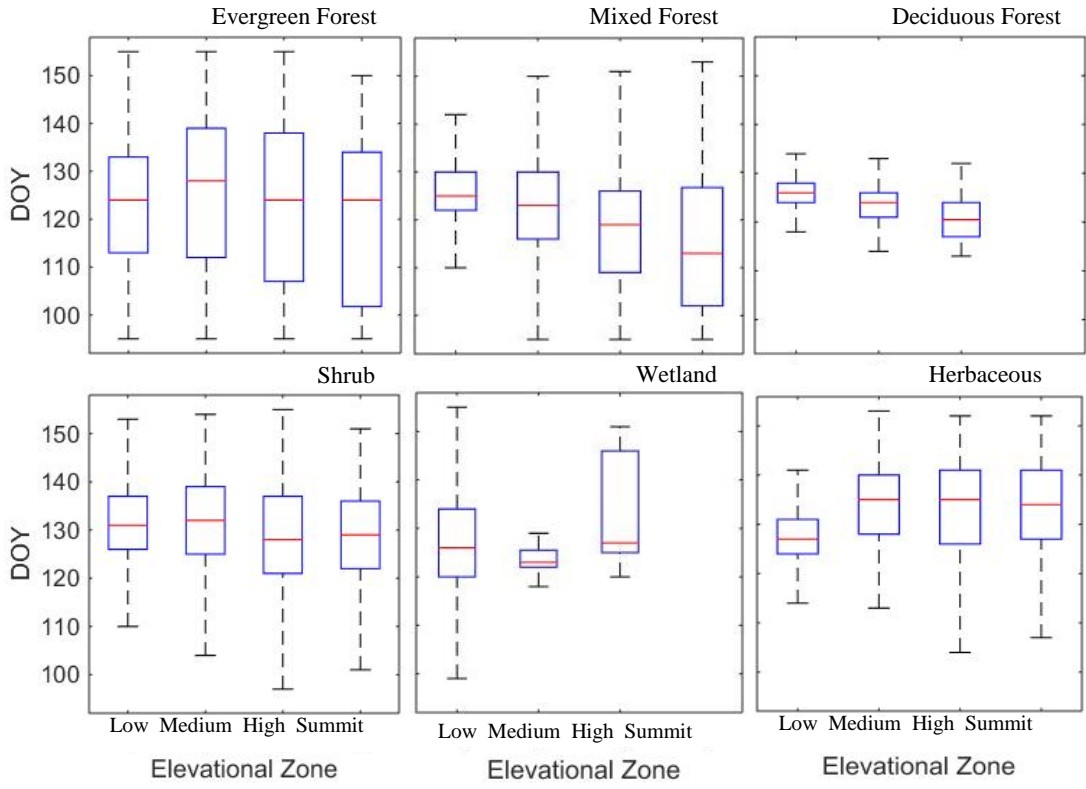

Figure 11. Greenup dates variation among different elevational zones for each landcover in 2013



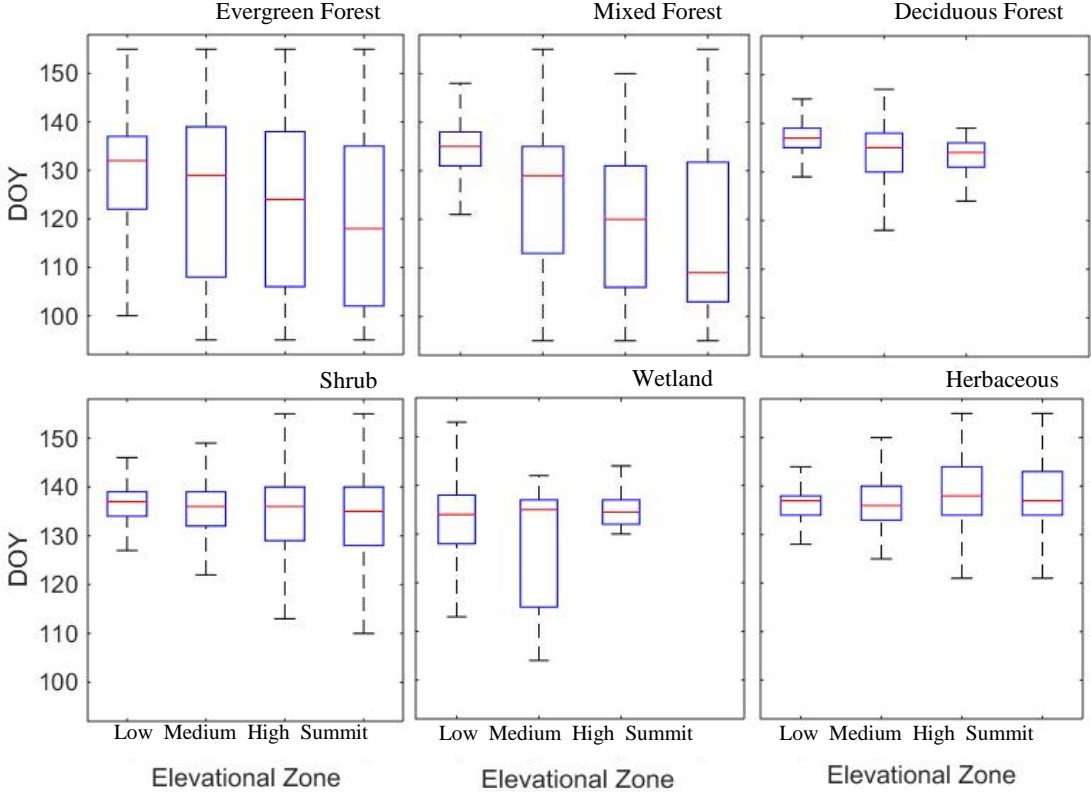

**Figure 12. Greenup dates variation among different elevational zones of each landcover in 2014**





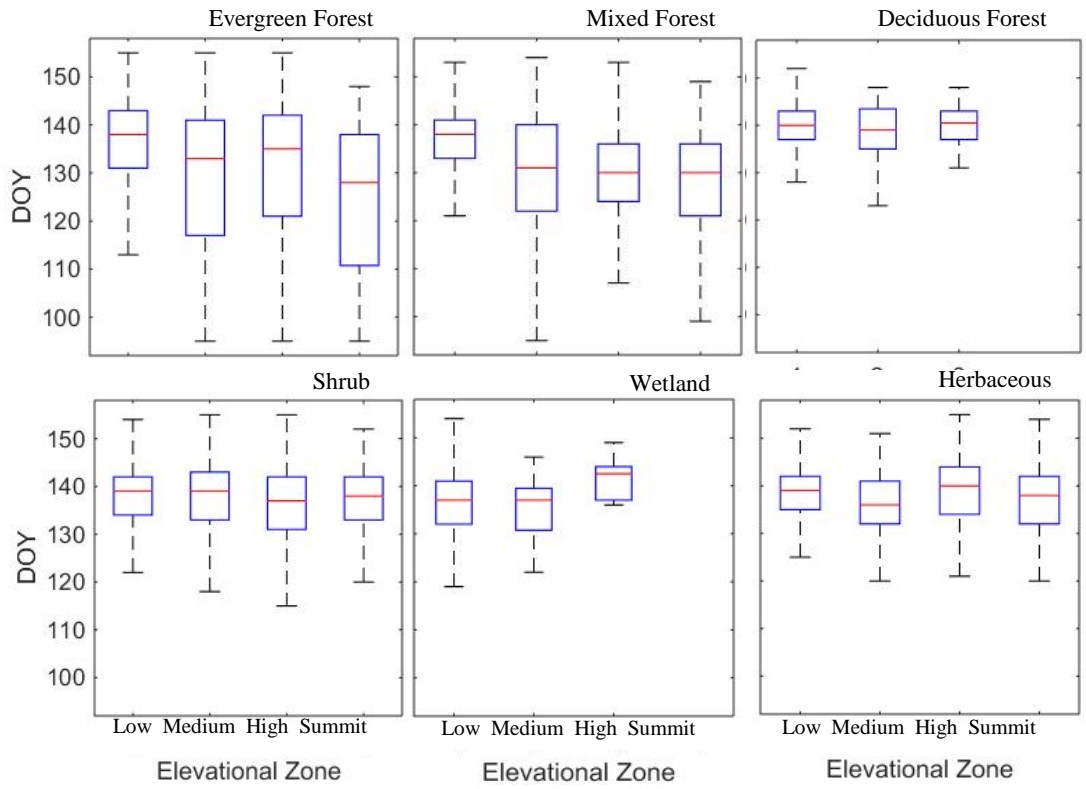

**Figure 13. Greenup dates variation among different elevational zones of each landcover in 2015**





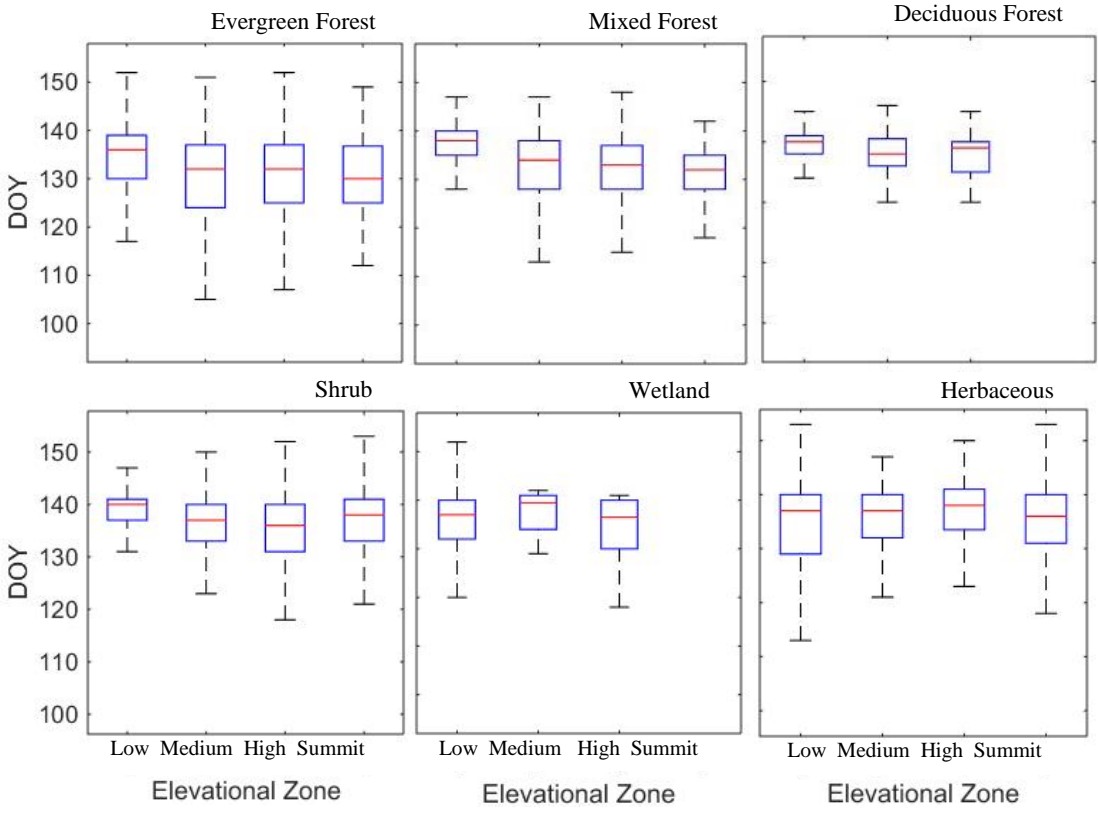

**Figure 14. Greenup dates variation among different elevational zones of each landcover in 2016**