# Peer review of "Using Remote Sensing to Monitor the Spring Phenology of Acadia National Park across Elevational Gradients"

_Biogeosciences, 2019_

## Short Comment (SC1) · 22 May 2019

General comments This is potentially a very interesting paper in which the authors compared satellite-derived greenup with direct observations of spring phenology across an elevational gradient in Acadia National Park, USA. The paper is generally well written, and the table and figures are generally useful, informative and well presents. It would be useful to have a clearer statement of the key message(s) and conclusion(s) from the study. Specific comments P1S30 The papers cited for the definition of 'spring phenology' are possibly not the original ones. Chmielwiski and Rotzer 2001 IJB and Menzel et al. 2006 GCB should be considered as citations for 'changes in phenology

are sensitive indicators of ecosystem response to climate change'. P2S5-10 Consider including some classic European references. Indicate whether in situ observations or satellite-derived phenology are being referred to. Ground observations may be laborious but they are also necessary to validate remotely sensed phenology. P2S15 The paper by Liu et al. 2016 IJB examining changes in greenup derived from EVI2 across different elevations on the Tibetan Plateau would be useful here and also in the discussion. P3S10 Would it be possible to provide hypotheses to test rather than the stated objectives and/or to indicate why the objectives are important. P3S15-20 Consider adding an insert of the USA in Figure 1 to indicate the location of Maine. Also consider if all the data in Figure 2 are necessary when the paper is focused on spring and consider combining Figures 2 and 3 as the information appears to be duplicated. Justify the use of March, April and May for use as representing spring temperature. How representative are the temperature data of the higher elevation areas? P4S15 Since this paper is focused on the spring season consider removing reference to maturity, senescence and dormancy. P5S5-10 It would be useful to have more detail on the 'thirty plant species'. Consider adding a table of the plant species, including number of individuals in each species, elevation range, functional type, frequency of observation, etc. If the direct observation data were collected by citizen scientists this could be stated. It would be useful to include how the elevational ranges used were derived. L3 indicates 'small trees' were monitored but later 'overstory deciduous trees' are referred to – please clarify which type of trees were monitored. Figure 4 is not very clear – the different elevation zones are not clearly depicted. P5S30 The species composition of the four 30m pixels include bunchberry but I am not entirely sure the scientific name matches the common name please check this. Also, all scientific names should be italicised. P6S5-15 A statistical analysis section could be included in 'Analysis of greenup dates' to include details of how the comparisons between different methods were made e.g. correlation? It would also be useful to see how the greenup and first leaf dates were compared to the temperature data. If the first leaf out dates of all the vegetation types are averaged this means that the later leafing species such as trees compared

to early leafing shrubs or ground vegetation will bias the 'true' green up date that is detected by the satellite data? It might be worth separating the field observations into different functional groups as it might turn out that early spring greenup is more closely related to ground-level than canopy level leaves. The method of averaging use would have introduced large variation. P6S30 Figures 6 and 7 are very interesting and clearly show different spatial scales. However, it would be useful to mention why the trends appear to be quite different. For example, in Figure 7 greenup appears to be earlier (more blue) in 2013 and later in 2014, 2015 and 2016 (more green, yellow and red) compared to Figure 6 (lower resolution). P7S5-10 Has the average May temperature been correlated with greenup or leaf out or is this just visual comparison? Average May temperature would be influenced strongly by daily temperature after leaf out – it is generally the temperature before the event that influences the timing. Are the temperature differences between years statistically significantly different? Have other temperature influences been examined? Is there any explanation why satellite derived greenup was later than field observations in 2016 and not in the other two years? P7S25-30 The boxplots for 2013 do not portray a convincing difference between low and high elevation, the range is very large and without any statistical analysis it is not accurate to state that earliest greenup dates were at lower elevation. The earlier greenup in 2015 (in Figure 8) is not so evident in Figure 10. P7S30-P8S5 Figures 11-14 (not referred to in the text) show interesting results – in general it appears that greenup was earlier at higher elevation for both deciduous and mixed forest which is contrary to that stated in the text. It is also interesting that deciduous forest greenup appears to be later than shrubs, wetland and herbaceous habitats which is what might be expected. P8-9 The discussion is a bit light and could include more references such as Liu et al. 2016 IJB. Perhaps discuss influence of temperature at different elevations and over different time periods, influence of moisture (especially at higher elevation), need for more field observations both species and years, and perhaps PhenoCams data, etc. It would be worth exploring the fact that since it appears that the variation for both direct observation and satellite-derived greenup are very large and so may explain some

of the correlation between the methods. Technical corrections Overall the manuscript could be improved by thorough editing for correct use of English. There were numerous minor, but important, errors, for example, use of 'medium' instead of 'median', use of 'doesn't' rather than 'does not' etc. Tenses should also be consistent either all in the past or present but not a combination of both. Use of symbols rather than words e.g. '°C' instead of 'Celsius degrees'. Also all figure captions require revision to include more detail. Alison Donnelly, University of Wisconsin-Milwaukee
* * *

---

## Referee Comment (RC2) · Anonymous Referee #2 · 12 Jun 2019

In "Using Remote Sensing to Monitor the Spring Phenology of Acadia National Park across Elevational Gradients", Liu et al. have presented a comparative study of greenup phenology of a mixed species ecosystem on a climate gradient. The data were collected and compared for four consecutive years (2013-2016) using satellite remote sensing and field observations along three hiking trails from low elevation to the summit of the mountains. The topic is relevant, and the manuscript is potentially interesting, however, I have several major concerns regarding the method and analysis of the study, the presentation of the results and the discussion:

**General Comments:**

1) The greenup data (from remote sensing) are quantitative, as offered by vegetation indices such as EVI and EVI2 (in this study) but the current manuscript is almost entirely based on qualitative comparison of the data from different sources and at across years. While the authors have gathered data from several independent sources, unfortunately most claims in the manuscript are not supported by numbers. I strongly recommend the authors to perform a comprehensive quantitative analysis to compare the data and support the discussion. There are few places that this can be improved, here is a few: a) evaluate the effect of spatial heterogeneity across scales, as it is claimed but not proved in the manuscript, b) evaluate control of climate drivers in the study area, for example using a simple phenology model at varying spatial scales. Also, it is not clear how the leaf-out date is extracted from raw citizen-science data into species-level and plot-level data and then into the satellite cell size aggregated data.

2) The discussion and conclusion sections are very short without proper reference to the results. Additionally, the novelty of the manuscript and the "take-home-message" is not clearly stated. It is expected to see the most significant message in the abstract, introduction and supported by the result and restated in the discussion. I believe the manuscript should be significantly improved in this regard. The method section does not say much about how the main research questions, mentioned at the end of intro, are answered and how the analyses are performed.

3) The authors have shown many figures in the manuscript but in order to make the paper easier to read, presentation of the results and figures must be improved. For example, many figures are not critically important for the discussion and the result. Figure 6 and 7 are showing the same quantity (greenup) from different sources but hard to compare and see the differences between Landsat and VIIRS. Or Figures 10 to 14 can be organized if merged into a single figure. These are only a few examples and unfortunately there are more places that the figures can be improved. Also figure captions do not include any message.

4) The writing of the paper is generally fine but it needs a proofread throughout. There are many places that should be revised/corrected. For example, there are several instances of use of contraction in the text that should be generally avoided. I'm a surprised to see repetitive contracted phrases in the manuscript such as: "it's", "doesn't" …..

**Special comments:**

- P1, L22: "Therefore, the greenup …" , How? Why? The sentence does not appear to be logically connected to previous statement.
- P2, L1: "… , and so on.". Vague term, should be avoided
- P3, L3: Abrupt transition after "…were used as well."
- P4, L20: use of MODIS-based EVI for Landsat is not properly justified.
- P5, L14: Evergreen phenology actually plays a role in the overall trends of the vegetation indices. Can the fraction of EN to DB trees be estimated from the NLCD dataset?
- P5, L29: "it's" should be avoided. Revise to "it is"
- P7, L20, two back to back parentheses seems awkward.
- P8, L13: "it's" should be avoided. Revise to "it is"
- P9, L13: "it's" should be avoided. Revise to "it is"
- P16, F1: "doesn't" should be avoided. Revise to "does not"
- P20, F6: "doesn't" should be avoided. Revise to "does not"
- P21, F7: "doesn't" should be avoided. Revise to "does not"
- Fig 2 is not super informative. The overall trend of temperature would be more visible if only Tmean is shown. Several symbols is just confusing.
- Fig 3: Again, not very informative figure. Not sure why the first two words in the caption are in the title-case. Months can be shown different symbols too, particularly for B/W print.

- Fig 4: Different elevation zones are supposedly shown in different colors (polygons?) but not visible in the map
- Fig 5: How is the bias calculated. A color-base is missing, caption is not sufficient to describe the color scale. What does the 0.025 offset line mean?
- Fig 6 and 7 should be combined for easier comparison. Maybe showing the difference map?. "The Park" should be "the park".
- Fig 8: What is the bias? Is the RMSE calculated only from three points? What's the unit?
- Fig 9: report R-squared.
- Fig 10: Not clear what the message of this figure is. what is the trend across years, or elevation? "Greenup" should be in lower-case.
- Fig 10-14, how many observations for each box plot? What do the whiskers and boxes show?

---

## Author Comment (AC1) · 18 Jun 2019

Dear Reviewer, Thank you so much for your time reading this paper and providing these valuable comments. Following are responses to your detailed comments. P1S30, P2S5-10, P3S10: Thank you for the suggestions of these papers. I'll incorporate these papers and your suggestions in the revised version. P3S10: Importance of the objectives will be added. P3S15-20: Map of USA indicating the location of Maine will be added. I agree there are duplicated information in figure 2 and 3. I can move figure 2 to supplementary material. The air temperature provided here is from the McFarland Hill weather station (elevation of 158 m) to show the yearly variation. Base on

the elevation, this station is in the low elevational zone. Caitlin McDonough MacKenzie, one of the coauthors, also measured temperature in each elevational using HOBO temperature loggers, and the results can be found in the supporting information of McDonough MacKenzie et al., 2019 (Table S1). From the results we can see that the maximum difference between the temperature of different elevational zone could even reach 2.4 degree (Cadillac ridge, 2016). Therefore, the temperature distribution in such topography complex region are quite heterogeneous. P4S15: maturity, senescence and dormancy will be removed from the paragraph P5S5-10: Details of the common species observed on the trail and their distribution in each elevational zone can be found in the supplement material of this paper (Table S3 and Table S4). Function type will be added in these tables in the revised version. The field study is led by my co-author Caitlin McDonough MacKenzie, and the results have been published in McDonough MacKenzie et al., 2019, which are also cited in the paper. As for the observation frequency, the trails were monitored twice a week from April 1 through June 30 as mentioned in P5S10-15. The tree monitored in the field are deciduous trees. In the field, only trees at eye height are observed. I'll change to "small deciduous trees" in L3. P5S30: Thank you for pointing out to italicize scientific names here. I'll make the changes in the revised version accordingly. Also double checked the scientific name, it is corrected. P6S5-15: I'll group Landsat monitored greenup dates and field observed leaf out dates base on their function types, then perform the comparison, and illustrate the variation of the dates for each group in figure 9. The comparison greenup and first leaf dates with the temperature of each elevational zone will be added in the revised version. P6S30: More detailed description and discussion regarding these two figures will be added in the revised version. P7S5-10: The statement here is drawn based on visual comparison. McDonough MacKenzie et al., 2019 found a significant linear relationship between mean spring temperatures and leaf out dates. Her finding will be cited in the revised version. A few reasons could lead to the difference between satellite monitored and field observed phenology, including variation in leaf out times between species, and the overall contribution of uncommon species (excluded from

our analysis) to green up as satellites capture the overall information of the pixel and field survey reflect individual plants. Based on the three years data, it's difficult to draw conclusion why satellite derived greenup was later than field observations in 2016 and not in the other two years. Maybe more years of analysis will help, which we'll perform in future study. P7S25-30: The conclusion of 2013 is based on the median value (the red line in the figure). More detailed statistical analysis, such as ANOVA, will be added in the revised version. The greenup in 2015 is actually later than other years in Figure 8 with more red and yellow. P7S30-P8S5: One possible reason for the greenup of forest earlier at higher elevation could be the differences in species composition of each landcover for each elevational zone. It's quite difficult to conduct detailed survey in this area due to the extreme topography variation. We'll conduct more years of analysis in the future to confirm this. P8-9: More discussion will be added in the revised version based on your above comments. There is PhenoCam installed in Acadia (https://phenocam.sr.unh.edu/webcam/sites/acadia/). It can be used to access the phenology yearly variation. However, it's difficulty to reflect the phenology variation along the elevation at this moment. Technical corrections: These errors will be corrected in the revised version.

Reference: McDonough MacKenzie, C., Primack, R. B. and Miller-Rushing, A. J.: Trails-as-transects: phenology monitoring across heterogeneous microclimates in Acadia National Park, Maine, Ecosphere, 10(3), e02626, doi:10.1002/ecs2.2626, 2019.

---

## Author Comment (AC3) · 5 Jul 2019

Dear Reviewer, Thank you so much for your time reading this paper and providing these valuable comments. Following are responses to your comments. 1)- a) The spatial heterogeneity evaluation of elevation will be added by showing histogram of elevation variance within 500 m pixels in research area using 30m ASTER DEM. The spatial heterogeneity of landcover will be evaluated using histograms showing the proportion of each landcover and numbers of landcovers within 500 m pixels. b) As temperature is the control climate drivers of this ecosystem, the comparison of greenup and first leaf out dates with the temperature of each elevational zone will be added in the revised

manuscript. The citizen data is collected in each elevational zone as mentioned in P5 L13-16. Therefore these data are considered as species level data in each elevational zone (plot-level data). To match with the field data, the satellite pixels were grouped in each elevational zone as mentioned in P6 L5-13. We do recognize that in this way "the species composition along the hiking trails may be quite different than the species composition of an entire Landsat pixel that covers and adjoins the trails" (P8L30-P9-L2). Therefore we tried to conduct species survey at Landsat scale along the hiking trail and a few relatively homogeneous were found (P5 L28-30). Comparison between the greenup of these pixels and the leaf out of the dormant species are performed as stated in P7 L2-14. In addition, RC1 suggested to "separating the field observations into different functional groups" for the comparison between field and satellite moni-tored phenology, which will also be added in the manuscript. 2) Discussion about the heterogeneity in elevation and land cover of this region, the relations between greenup and temperature of each elevational zone will be added and reflected in conclusion. Also the current discussion will be extended as well, such as need for more field ob-servations and longer satellite monitoring. The novelty of this manuscript is to monitor phenology in mountainous region at 30 m scale to reflect the detailed variation in this heterogeneous region, and the take-home-message is 30 m is a better scale for phe-nology monitoring and agree well with field monitored leaf out dates. The manuscript will be revised to highlight these points. The main research questions will be stated at the beginning of the method section with an overall analyze plan followed. 3) Figure 6 and 7 will be merged into one figure with Landsat results and VIIRS results lie together for better comparison. Figure 10 to 14 will be grouped into a single figure with different year's box plot in different colors in one plot, and each plot shows the information of one landcover type. Figure 2 will be moved to supplementary material and only mean temperature will be displayed. Figure 4 will be enlarged and the trails will be colored differently base on the elevational zones. The variance of greenup of each elevational zone will be add in figure 9. And more information will be provided in the captions. 4) Corresponding corrections will be made in the manuscript. Special comments: P1L22,

the sentence will be changed to "Therefore, to take advantages of the spatial resolution of 30 m data and the temporal resolution of 500 m data, the Landsat 8 Enhanced Vegetation Index (EVI) data was augmented with Spatial and Temporal Adaptive Reflectance Fusion Model (STARFM) simulated EVI values to monitor the greenup of this mountainous region, and it does provide more spatial details than VIIRS data alone and agree well with field monitored leaf out dates. P2L1, the sentence will be changed to "such changes provide important feedbacks to the climate system, such as through albedo, nutrient cycling" P3L3, the sentence will change to "MODIS daily Nadir Bidirectional Reflectance Distribution Function Adjusted Reflectance (MCD43, V006) were used together with Landsat 8 surface reflectance to generate simulated 30 m images to improve the temporal resolution of 30 m." P4L20, MODIS EVI was used to simulate 30 m images as mentioned in P4L26-28. P5L14, The comparison of satellite monitored greenup dates with field observed leaf out dates was performed at 30 m scale (P6L5-L13). The landcover map used in the manuscript is 30 m as well (P6L10-13), which classify evergreen forest. However, the fraction of EN to DB trees of each 30 m trees are provided. P5L29, P8L13, P9L13, P16FQ, P20F6, P21F7, corresponding changes will be made in the manuscript. P7L20: the sentence will be changed to "with a small bias (-2 days) and RMSE (10 days) as displayed in Fig.9. " Fig 2. This figure will be moved to supplement material as suggested by RC1. Fig 3. corresponding changes will be made in the manuscript.